

# GMPR: A robust normalization method for zero-inflated count data with application to microbiome sequencing data

Li Chen[1], James Reeve[2], Lujun Zhang[3], Shengbing Huang[2], Xuefeng Wang[4] and Jun Chen[2,5]

[1] Department of Health Outcomes Research and Policy, Harrison School of Pharmacy, Auburn University, Auburn, AL, USA
[2] Bioinformatics and Computational Biology Program, University of Minnesota—Rochester, Rochester, MN, USA
[3] College of Environmental and Resource Sciences, Zhejiang University, Hangzhou, Zhejiang, China
[4] Department of Biostatistics and Bioinformatics, Moffitt Cancer Center, Tampa, FL, USA
[5] Division of Biomedical Statistics and Informatics and Center for Individualized Medicine, Mayo Clinic, Rochester, MN, USA

## ABSTRACT

Normalization is the first critical step in microbiome sequencing data analysis used to account for variable library sizes. Current RNA-Seq based normalization methods that have been adapted for microbiome data fail to consider the unique characteristics of microbiome data, which contain a vast number of zeros due to the physical absence or under-sampling of the microbes. Normalization methods that specifically address the zero-inflation remain largely undeveloped. Here we propose geometric mean of pairwise ratios—a simple but effective normalization method—for zero-inflated sequencing data such as microbiome data. Simulation studies and real datasets analyses demonstrate that the proposed method is more robust than competing methods, leading to more powerful detection of differentially abundant taxa and higher reproducibility of the relative abundances of taxa.

## INTRODUCTION

High-throughput sequencing experiments such as RNA-Seq and microbiome sequencing are now routinely employed to interrogate the biological systems at the genome scale (*Wang, Gerstein & Snyder, 2009*). After processing of the raw sequence reads, the sequencing data usually presents as a count table of detected features. The complex processes involved in the sequencing causes the sequencing depth (library size) to vary across samples, sometimes ranging several orders of magnitude. Normalization, which aims to correct or reduce the bias introduced by variable library sizes, is an essential preprocessing step before any downstream statistical analyses for high-throughput sequencing experiments (*Dillies et al., 2013*; *Li et al., 2015*). Normalization is especially critical when the library size is a confounding factor that correlates with the variable of interest. An inappropriate normalization method may either reduce statistical power with

Corresponding author
Jun Chen, chen.jun2@mayo.edu

the introduction of unwanted variation, or more severely, result in falsely discovered features. One popular approach for normalizing the sequencing data involves calculating a size factor for each sample as an estimate of the library size. The size factors can be used to divide the read counts to produce normalized data (in the form of relative abundances), or to be included as offsets in count-based regression models such as DESeq2 (*Love, Huber & Anders, 2014*) and edgeR (*Robinson, McCarthy & Smyth, 2010*) for differential feature analysis. One simple normalization method is total sum scaling (TSS), which uses the total read count for each sample as the size factor. However, there are a couple undesirable properties for TSS. First, it is not robust to outliers, which are disproportionately large counts that do not reflect the underlying true abundance. Outliers have frequently been observed in sequencing samples due to technical artifacts such as preferential amplification by PCR (*Aird et al., 2011*). Several outliers could lead to the overestimation of the library size if not properly addressed. Second, it creates compositional effects: non-differential features will appear to be differential due to the constant-sum constraint (*Tsilimigras & Fodor, 2016*; *Mandal et al., 2015*; *Morton et al., 2017*). Compositional effects are much stronger when the differential features are highly abundant or their effects are in the same direction (not balanced). An ideal normalization method should thus capture the invariant part of the count distribution and be robust to outliers and differential features.

Many normalization methods have been developed for sequencing data generally, and for RNA-Seq data in particular (*Dillies et al., 2013*; *Li et al., 2015*). These methods mostly rely on the assumption that the dataset to be normalized has a large invariant part and the majority of features do not change with respect to the condition under study. Robust statistics such as median and trimmed mean, which are not sensitive to a small set of differential features, are frequently used to estimate the library size. Two popular normalization methods for RNA-Seq data include trimmed mean of M values (TMM, implemented in edgeR) (*Robinson & Oshlack, 2010*) and the DESeq normalization (equivalent to relative log expression normalization implemented in edgeR. For simplicity, we label it as "RLE.") (*Anders & Huber, 2010*). RLE method calculates the geometric means of all features as a "reference," and all samples are compared to the "reference" to produce ratios (fold changes) for all features. The median ratio is then taken to be the RLE size factor. TMM method, on the other hand, selects a reference sample first, and all other samples are compared to the reference sample. The trimmed (weighted) mean of the log ratios is then calculated as the TMM size factor (log scale). Compared to RNA-Seq data, microbiome sequencing data are more over-dispersed and contain a vast number of zeros. For example, the human fecal microbiome data set from a study of the long-term dietary effect on the gut microbiota ("COMBO" data) contains 1,873 non-singleton operational taxonomic units (OTUs, a proxy for bacterial species) from 99 subjects and more than 80% of the OTU counts are zeros (*Wu et al., 2011*). Excessive zeros lead to a small number of "core" OTUs that are shared across samples. For the COMBO dataset, none of the OTUs are shared by all samples and only five OTUs are shared by more than 90% samples. For RLE, the geometric means of OTUs are not well defined for OTUs with 0s, and OTUs with 0s are typically excluded in size factor

calculation. We are thus left with a very small number of common OTUs to calculate the size factor. As the OTU data become more sparse, RLE becomes less stable. For datasets like COMBO data, where there are no common OTUs, RLE fails. For TMM, a reference sample has to be selected before the size factor calculation. Reliance on a reference sample restricts the size factor calculation to a specific OTU set that the reference sample harbors (77–433 OTUs for COMBO data). Therefore, both RLE and TMM use only a small fraction of the data available in the OTU data and are not optimal from an information perspective.

One popular strategy to circumvent the zero-inflation problem is to add a pseudo-count (*Mandal et al., 2015*). This practice has a Bayesian explanation and implicitly assumes that all the zeros are due to under-sampling (*McMurdie & Holmes, 2014*). However, this assumption may not be appropriate due to the large extent of structural zeros due to physical absence. Moreover, the choice of the pseudo-count is very arbitrary and it has been shown that the clustering results can be highly dependent upon the choice (*Costea et al., 2014*). Recently, a new normalization method cumulative sum scaling (CSS) has been developed for microbiome sequencing data (*Paulson et al., 2013*). In CSS, raw counts are divided by the cumulative sum of counts, up to a percentile determined using a data-driven approach. The percentile is aimed to capture the relatively invariant count distribution for a dataset. However, the determination of the percentiles could fail for microbiome datasets that have high count variability. Therefore, a more robust method to address the zero-inflated sequencing data is still needed.

Here we propose a novel inter-sample normalization method geometric mean of pairwise ratios (GMPR), developed specifically for zero-inflated sequencing data such as microbiome sequencing data. By comprehensive tests on simulated and real datasets, we show that GMPR outperforms the other competing methods for zero-inflated count data.

## METHODS

### GMPR normalization details

Our method extends the idea of RLE normalization for RNA-Seq data and relies on the same assumption that there is a large invariant part in the count data. Assume we have a count table of OTUs from 16S rDNA targeted microbiome sequencing. Denote the $c_{ki}$ as the count of the $k$th OTU ($k = 1, \ldots, q$) in the $i$th ($i = 1, \ldots, n$) sample. The RLE method calculates the size factor $s_i$, which estimates the (relative) library size of a given sample, based on

- Step 1: Calculate the geometric means for all OTUs

$$\mu_k^{\mathrm{GM}} = (c_{k1} c_{k2} \ldots c_{kn})^{1/n}, \; k = 1, \ldots, q$$

- Step 2: For a given sample,

$$s_i = \mathrm{median}_k \left\{ c_{ki} / \mu_k^{\mathrm{GM}} \right\}, \; i = 1, \ldots, n$$

Since geometric mean is not well defined for features with 0s, features with 0s are usually excluded in size calculation. However, for zero-inflated data such as microbiome sequencing data, as the sample size increases, the probability of existence of features without any 0s becomes smaller. It is not uncommon that a large dataset does not share any common taxa. In such cases, RLE fails. As an alternative, a pseudo-count such as 1 or 0.5 has been suggested to add to the original counts to eliminate 0s (*Mandal et al., 2015*). Since the majority of the counts may be 0s for microbiome data, adding even a small pseudo-count could have a dramatic effect on the geometric means of most OTUs. To circumvent the problem, GMPR reverses the order of the two steps of RLE. The first step is to calculate $r_{ij}$, which is the median count ratio of nonzero counts between sample $i$ and $j$,

$$r_{ij} = \underset{k \in \{1, \ldots, q\} \mid c_{ki} \cdot c_{kj} \neq 0}{\text{Median}} \left\{ \frac{c_{ki}}{c_{kj}} \right\},$$

The second step is to calculate the size factor $s_i$ for a given sample $i$ as

$$s_i = \left( \prod_{j=1}^{n} r_{ij} \right)^{1/n}, i = 1, \ldots, n.$$

Figure 1 illustrates the procedure of GMPR. The basic strategy of GMPR is that we conduct the pairwise comparison first and then combine the pairwise results to obtain the final estimate. Although only a small number of OTUs (or none) are shared across all samples due to severe zero-inflation, for every pair of samples, they usually share many OTUs. For example, 83 OTUs are shared on average for COMBO sample pairs. Thus, for pairwise comparison, we focus on these common OTUs that are observed in both samples to have a reliable inference of the abundance ratio between samples. We then synthesize the pairwise abundance ratios using a geometric mean to obtain the size factor. Based on this pair analysis strategy, we utilize far more information than RLE and TMM, both of which are restricted to a small subset of OTUs. It should be noted that GMPR is a general method, which could be applied to any type of sequencing data in principle.

The R implementation of GMPR could be accessed by https://github.com/jchen1981/GMPR.

## Simulation studies to evaluate the performance of GMPR normalization

We study the performance of GMPR using simulated OTU datasets. Specifically, we study the robustness of GMPR to differential and outlier OTUs, and the effect on the performance of differential abundance analysis (DAA) of OTU data. We compare GMPR to competing normalization methods including CSS, RLE, RLE with pseudo-count 1 (RLE+), TMM, TMM with pseudo-count 1 (TMM+) and TSS. The details of calculating the size factors using each normalization method are described in Table 1. The size factors from different normalization methods are further divided by the median so that they are on the same scale.

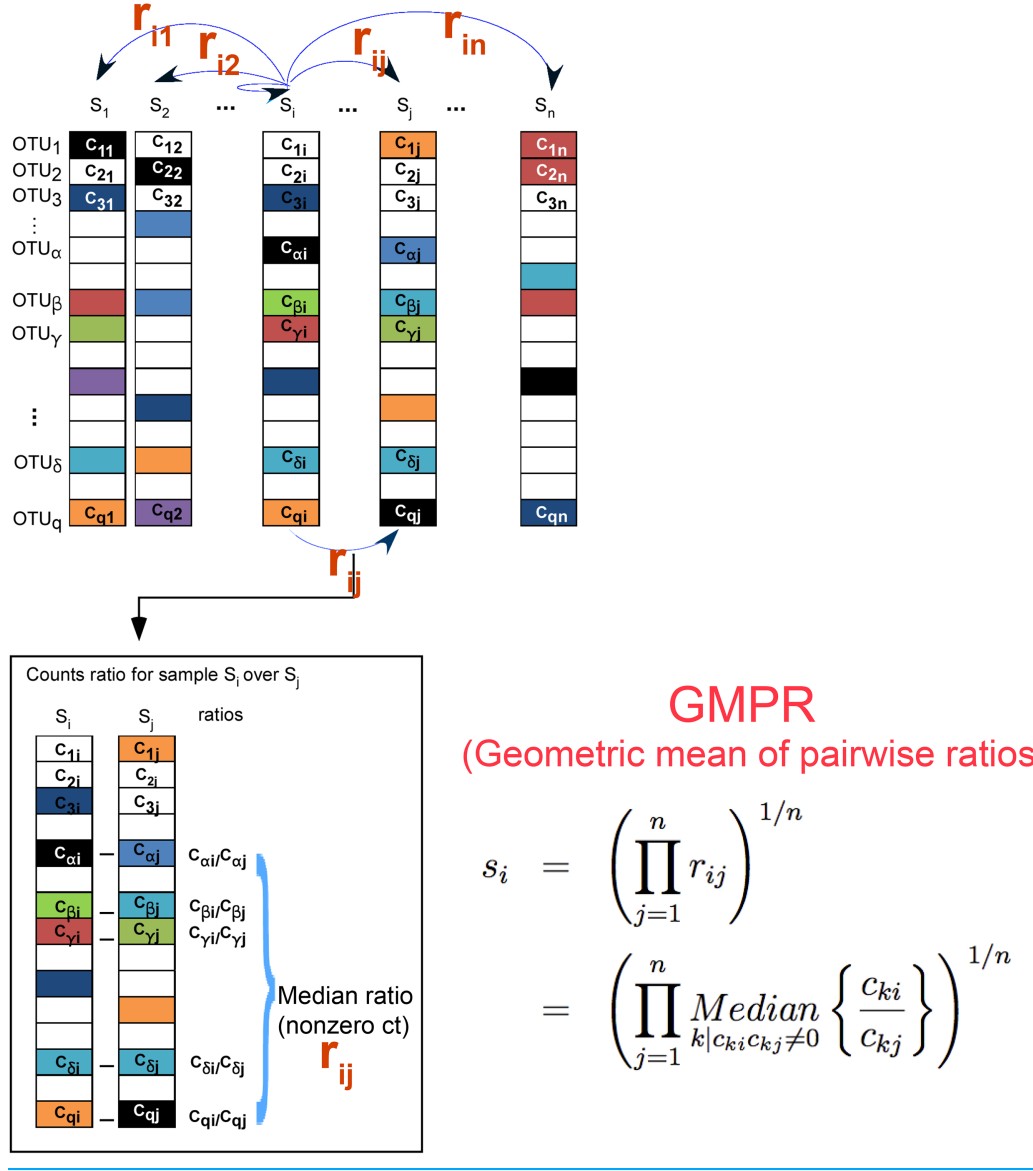

**Figure 1 GMPR starts with pairwise comparisons (upper). Each pairwise comparison calculates the median abundance ratio of those common OTUs between the pair of samples (lower).** The pairwise ratios are then synthesized into a final estimate.

### Robustness to differential and outlier OTUs

We first use a perturbation-based simulation approach to evaluate the performance of normalization methods, focusing on their robustness to differentially abundant OTUs and sample-specific outlier OTUs. The idea is that we first simulate the counts from a common probabilistic distribution so that the total count is a proxy of the "true" library size. Next, we perturb the counts in different ways and apply different normalization methods on the perturbed counts and evaluate the performance based on the correlation between estimated size factor and "true" library size. Specifically, we generate zero-inflated count data based on a Dirichlet-multinomial model with known

**Table 1 Calculation of size factors for normalization methods compared in the analysis.**

- GMPR (Geometric Mean of Pairwise Ratios): The size factors for all samples are calculated by GMPR described in the Method section.

- CSS (Cumulative Sum Scaling): The size factors for all samples are calculated by applying `newMRexperiment`, `cumNorm` and `normFactors` in Bioconductor package metagenomeSeq (*Paulson et al., 2013*).

- RLE (Relative Log Expression): The size factors for all samples are calculated by the `calcNormFactors` with the parameter set as "RLE" in the edgeR Bioconductor package (*Anders & Huber, 2010*). The scaled size factors are obtained by multiplying the size factors with the total read count.

- RLE+ (Relative Log Expression plus pseudo-counts): The scaled size factors for all samples are calculated in the same way as RLE, except that each data entry is added with a pseudo-count 1.

- TMM (Trimmed Mean of M values): The size factors for all samples are calculated by the `calcNormFactors` function with the parameter set as "TMM" in the edgeR Bioconductor package (*Robinson & Oshlack, 2010*). The scaled size factors are obtained by multiplying the size factors with the total read count.

- TMM+ (Trimmed Mean of M values plus pseudo-counts): The scaled size factors for all sample are calculated in the same way as TMM, except that each data entry is added with a pseudo-count 1.

- TSS (Total Sum Scaling): The size factors are taken to be the total read counts.

library sizes (*Chen & Li, 2013*). The mean and dispersion parameters of Dirichlet-multinomial distribution are estimated from the COMBO dataset after filtering out rare OTUs with less than 10 reads and discarding samples with less than 1,000 reads ($n = 98$, $q = 625$) (*Wu et al., 2011*). The library sizes are also drawn from those of the COMBO data. To investigate the effect of sparsity (the number of zeros), OTU counts are simulated with different zero percentages (~60%, 70% and 80%) by adjusting the dispersion parameter. A varying percentage of OTUs (0%, 1%, 2%, 4%, 8%, 16%, 32%, 64%) are perturbed in each set of simulation, with varying strength of perturbation. The counts $c_{ki}$ of perturbed OTUs are changed to $\sqrt{c_{ki}}$ or $c_{ki}^2$ for strong perturbation and $0.25c_{ki}$ or $4c_{ki}$ for moderate perturbation.

We employ two perturbation approaches where we decrease/increase the abundances of a "fixed" or "random" set of OTUs. As shown in Fig. 2, in the "fixed" perturbation approach, the same set of OTUs are decreased/increased in the same direction for all samples, reflecting differentially abundant OTUs under a certain condition such as disease state. In the "random" perturbation approach, each sample has a random set of OTUs perturbed with a random direction, mimicking the sample-specific outliers.

Finally, size factors for all methods are estimated and the Pearson's correlation between the estimated and "true" library sizes is calculated. The simulation is repeated 25 times and the mean estimate and its 95% confidence intervals (CIs) are reported.

### Effect on the performance of DAA

One use of the estimated size factor is for DAA of OTU data, where the size factor (usually on a log scale) is included as an offset in a count-based parametric model to address variable library sizes. Many count-based models have been proposed for DAA including

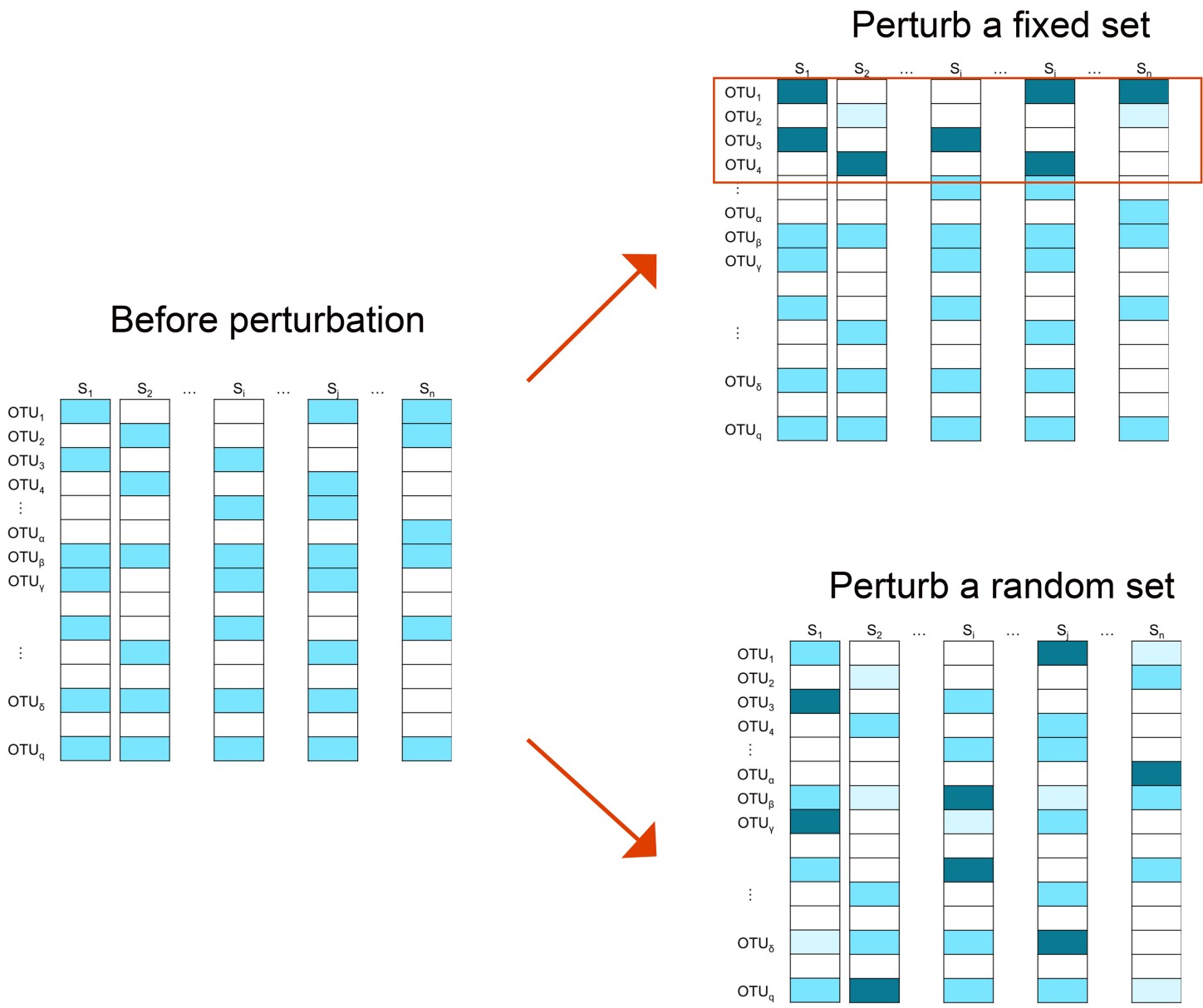

**Figure 2 Illustration of the simulation strategy.** In the "fixed" perturbation approach, the abundances of the same set of OTUs are decreased/increased for all samples, reflecting differentially abundant OTUs under certain conditions such as disease state. In the "random" perturbation approach, each sample has a random set of OTUs perturbed with a random direction, reflecting the sample-specific outliers. The darkness of the color indicates the OTU abundance.

DESeq2 and edgeR (*McMurdie & Holmes, 2014*). These methods usually come with their native normalization schemes such as RLE for DESeq2 and TMM for edgeR. Therefore, it is interesting to see if the GMPR normalization could improve the performance of these methods. To achieve this end, we use DESeq2 to perform DAA on the OTU table since DESeq2 has been shown to be more robust than edgeR for zero-inflated dataset (*Chen et al., 2018*). We compare the performance of DESeq2 using its native RLE normalization to that using GMPR or TSS normalization.

We use the same simulation strategy described in *Chen et al. (2018)*. Specifically, zero-inflated negative binomial distribution (ZINB) is used to simulate the OTU count data. ZINB has the following probability distribution function

$$f_{zinb}(c_{ki}|p_{ki}, \mu_{ki}, \phi_{ki}) = p_{ki} \cdot I_0(c_{ki}) + (1 - p_{ki}) \cdot f_{nb}(c_{ki}|\mu_{ki}, \phi_{ki}), \tag{1}$$

which is a mixture of a point mass at zero ($I_0$) and a negative binomial ($f_{nb}$) distribution of the form

$$f_{nb}(c_{ki}|\mu_{ki}, \phi_{ki}) = \frac{\Gamma\left(c_{ki} + \frac{1}{\phi_{ki}}\right)}{\Gamma(c_{ki} + 1)\,\Gamma\left(\frac{1}{\phi_{ki}}\right)} \cdot \left(\frac{\phi_{ki}\mu_{ki}}{1 + \phi_{ki}\mu_{ki}}\right)^{c_{ki}} \cdot \left(\frac{1}{1 + \phi_{ki}\mu_{ki}}\right)^{\frac{1}{\phi_{ki}}} \tag{2}$$

The three parameters—prevalence ($p_{ki}$), abundance ($\mu_{ki}$) and dispersion ($\phi_{ki}$)—fully capture the zero-inflated and dispersed count data. We generate the simulated datasets (two sample groups of size 49 each) based on the parameter values estimated from the COMBO dataset. Five percent of OTUs are randomly selected to have their counts in one group multiplied by a factor of four. The groups in which this occurs are randomly selected and thus the abundance change is relatively "balanced." To further study the performance under strong compositional effects, on top of the "balanced" simulation, we also select two highly abundant OTUs ($\pi = 0.168$ and $0.083$, respectively) to be differentially abundant in one group. We then apply DESeq2 on the simulated datasets with RLE, GMPR and TSS normalization, where we denote DESeq2-GMPR, DESeq2-RLE, DESeq2-TSS for these three approaches. For each approach, the *P*-values are calculated for each OTU and corrected for multiple testing using false discovery rate (FDR) control (Benjamini–Hochberg procedure). We evaluate the performance based on FDR control and ROC analysis, where the true positive rate is plotted against false positive rate at different *P*-value cutoffs. The observed FDR is calculated as

$$\frac{FP}{\max(1, \; FP + TP)}$$

where FP and TP are the number of false and true positives respectively. Simulation results are averaged over 100 repetitions.

## RESULTS

### Simulation: GMPR is robust to differential and outlier OTUs

We first study the robustness of GMPR to differentially abundant OTUs and sample-specific outlier OTUs by using the perturbation-based simulation approach, where we artificially alter the abundances of a "fixed" or "random" set of OTUs under different levels of zero-inflation, percentage of perturbed OTUs and strength of perturbation.

In the simulation of "fixed" perturbation (Fig. 3), the performance of all methods decrease in most cases with the increased zero percentage. TSS has excellent performance under moderate perturbation but performs unstably under strong perturbation (the correlation decreases steeply when the percentage of perturbed OTUs increases from 1% to 4%; after that, the correlation increases since the total sum moves closer to $\sum_k c_{ki}^2$,

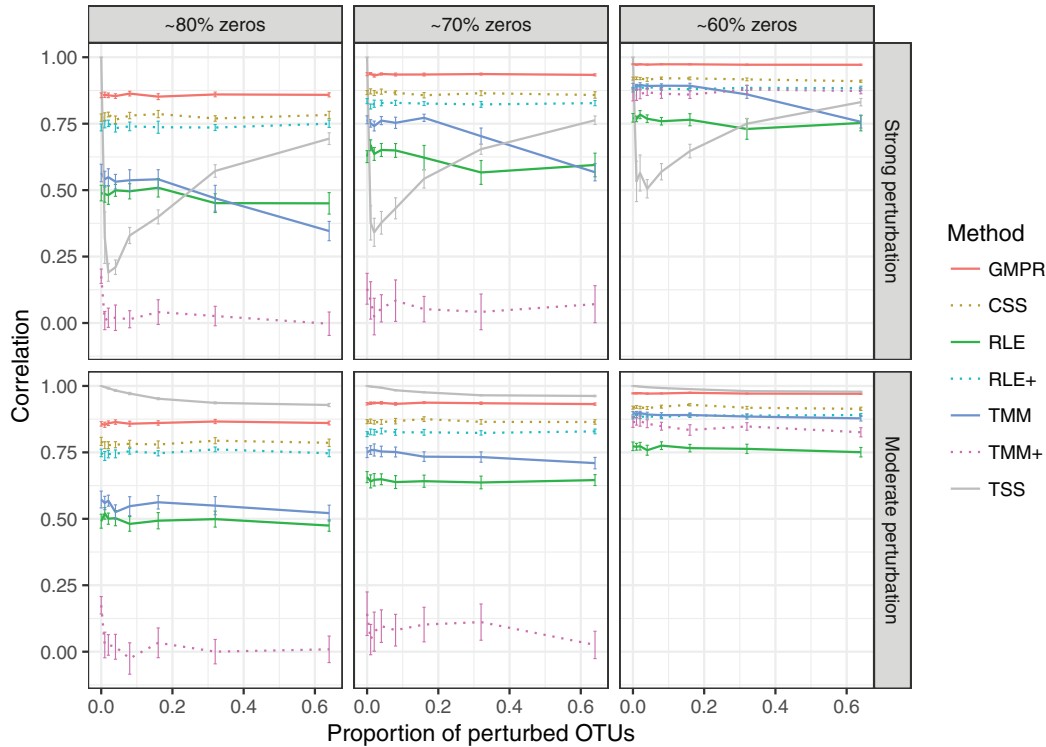

**Figure 3 Spearman's correlation between the estimated size factors and the simulated "true" library sizes when a fixed set of OTUs are perturbed.** The performance of different normalization methods are compared under different levels of zero-inflation, percentage of perturbed OTUs and strength of perturbation. Error bars represent 95% CIs.

which is highly correlated with the original library size $\Sigma_k c_{ki}$). GMPR, followed by CSS, consistently outperforms the other methods when the perturbation is strong. When the perturbation is moderate, GMPR is only secondary to TSS when the percentage of zeros is high (80%) and on par with TSS when the percentage of zeros is moderate (70%) or low (60%). For RNA-Seq based methods, TMM performs better than RLE in either strong or moderate perturbation. Though the performance of RLE+ improves by adding pseudo-counts to the OTU data, the size factor estimated by TMM+ merely correlates with true library size when the zero percentage is high (70% and 80%). In contrast, GMPR, together with CSS, performs stable in all cases and GMPR yields better size factor estimate than CSS.

In the "random" perturbation scenario (Fig. 4), performance of all methods decreases with the increased zero percentage as the "fixed" scenario. Similar to the performance in "fixed" perturbation scenario, TSS has excellent performance under moderate perturbation but performs poorly under strong perturbation. When the perturbation is strong, GMPR, followed by CSS, still outperforms the other methods. RNA-Seq based methods including TMM, TMM+, RLE and RLE+ have a similar trend as in "fixed" perturbation. However, compared to "fixed" perturbation, the performance of TMM and RLE decreases more obviously as the number of perturbed OTUs increases.

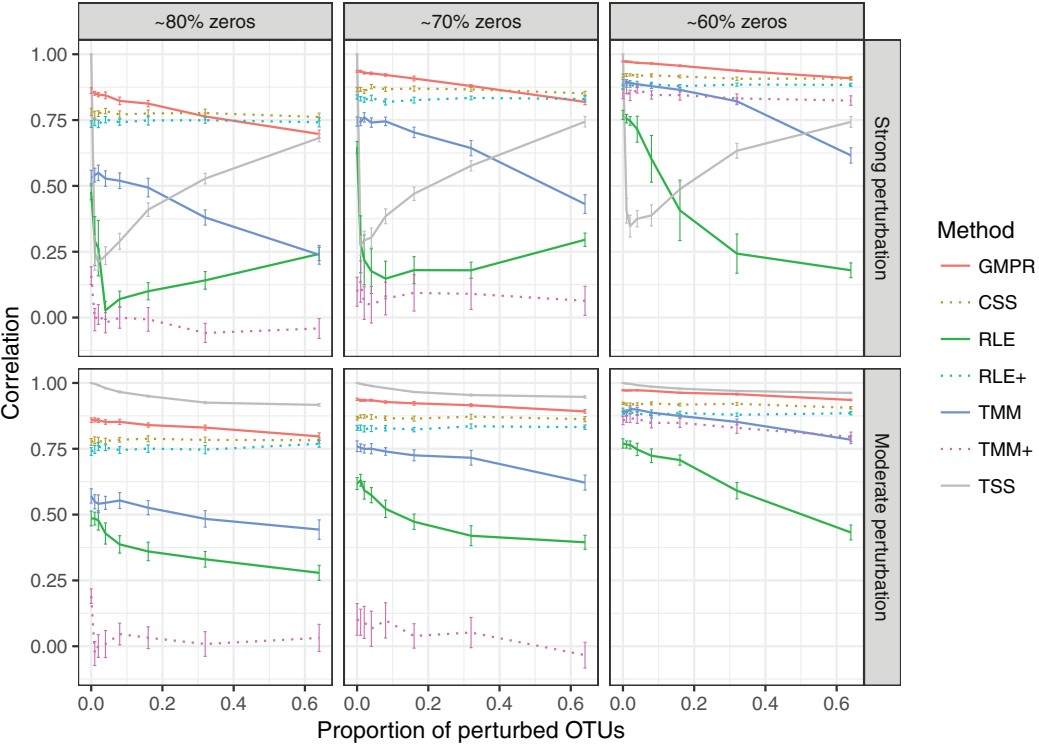

**Figure 4 Spearman's correlation between the estimated size factors and the simulated "true" library sizes when a random set of OTUs are perturbed.** The performance of different normalization methods are compared under different levels of zero-inflation, percentage of perturbed OTUs and strength of perturbation. Error bars represent 95% CIs.

In contrast, GMPR and CSS are more robust to sample-specific outlier OTUs in all cases and GMPR results in better size factor estimate than CSS.

## Simulation: GMPR improves the performance of DAA

In the previous section, we demonstrate that GMPR could better recover the "true" library size in presence of differentially abundant OTUs or sample-specific outlier OTUs. In this section, with a different perspective, we show that the robustness of GMPR method translates into a better false positive control and higher statistical power in the context of DAA, where the aim is to detect differentially abundant OTUs between two sample groups.

We simulate the zero-inflated count data using ZINB model and use DESeq2 to perform DAA with different normalization schemes (RLE, GMPR and TSS). In one scenario, we randomly select 5% OTUs to be differential with a fold change of four in either sample group (Scene 1). In the other scenario, in addition to the 5% randomly selected OTUs, we select two highly abundant OTUs to be differentially abundant in one group to create strong compositional effects (Scene 2). In this scenario, the abundance change of these highly abundant OTUs will lead to the change of the "relative" abundances of other OTUs if the TSS normalization is used. The results for the two scenarios are presented in Fig. 5. In Scene 1 (Figs. 5A and 5B), we see that all approaches have slightly elevated FDRs relative to the nominal levels (Fig. 5A), probably due to inaccurate *P*-value

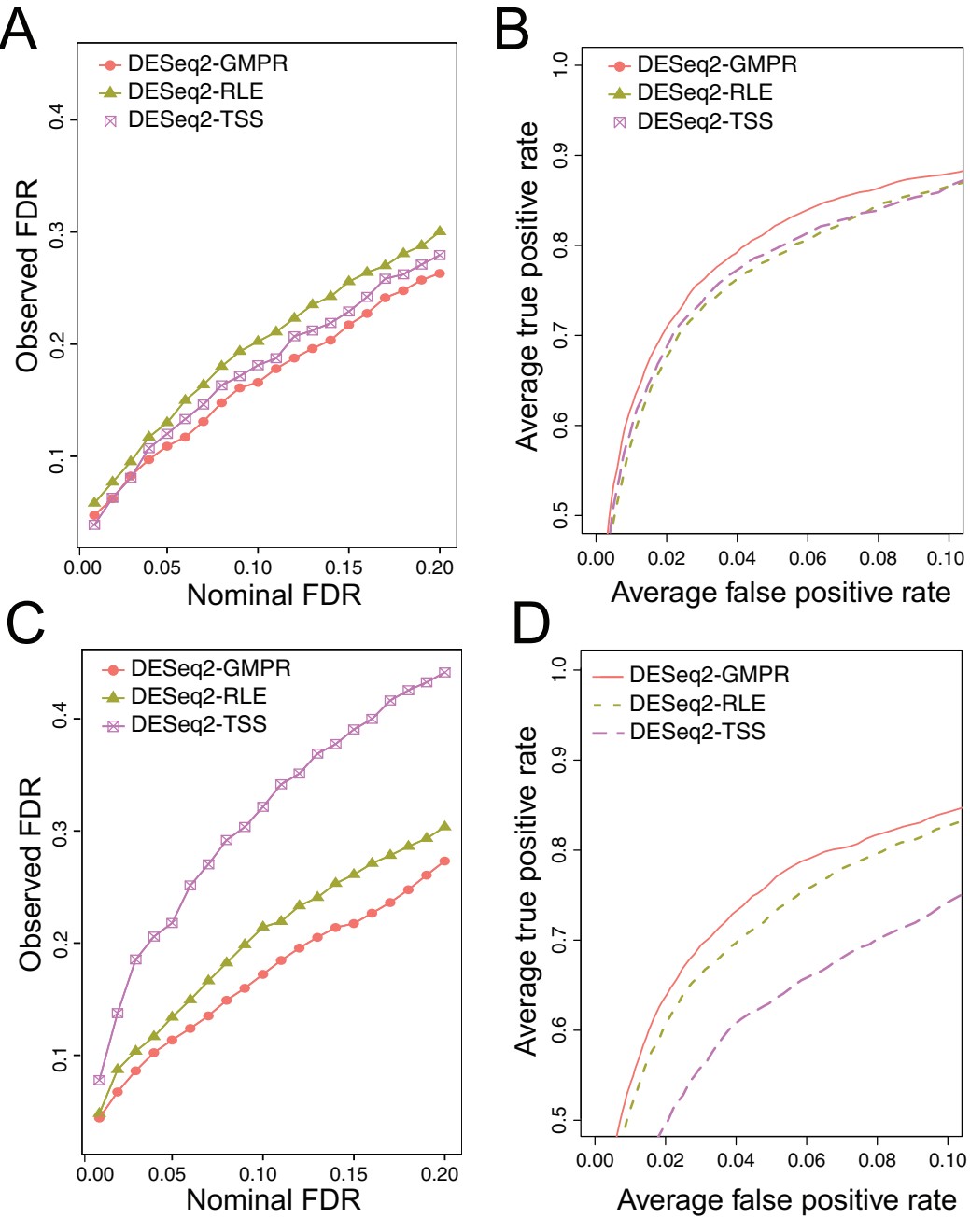

**Figure 5 Comparison of the performance of different normalization schemes in DESeq2-based differential abundance analysis.** (A, B): Scene 1 ("balanced" scenario), 5% random OTUs are differentially abundant between two groups with a fold change of four. (C, D): Scene 2 ("unbalanced" scenario), in addition to 5% random OTUs, two highly abundant OTUs are differentially abundant in one group to create strong compositional effects. (A, C): ability to control the FDR. The observed FDR is plotted against the nominal FDR level. (B, D): ROC curves to compare the power. The true positive rate is plotted against false positive rate at different *P*-value cutoffs.

calculation based on the asymptotic distribution of Wald statistic for those taxa with excessive zeros. Nevertheless, the observed FDR of DESeq2 using GMPR is closer to the nominal level than that using RLE (native normalization) and TSS. In terms of ROC-based power analysis (Fig. 5B), GMPR achieves a higher area under the curve than RLE and TSS. In this "balanced" scenario, TSS performs relatively well and is even slightly better than RLE. The performance differences are more revealing in Scene 2 (Figs. 5C and 5D), where we artificially alter the abundances of two highly abundant OTUs. In this setting, TSS has a poor FDR control due to strong compositional effects and has a much lower statistical power at the same false positive rate. In contrast, the performance of GMPR and RLE remains stable, and GMPR performs better than RLE in terms of both FDR control and power.

## Real data: GMPR reduces the inter-sample variability of normalized abundances

We next evaluate various normalization methods using 38 gut microbiome datasets from 16S rDNA sequencing of stool samples (Table S1). These experimental datasets were retrieved from Qiita database (https://qiita.ucsd.edu/) with a sample size larger than 50 each. The 38 datasets come from different species as well as a wide range of biological conditions. If a study involves multiple species, we include samples from the predominant species. We focus the analysis on gut microbiome samples because the gut microbiome is more studied than that from other sample types.

For the real data, it is not feasible to calculate the correlation between estimated size factors and "true" library sizes as done for simulations. As an alternative, we use the inter-sample variability as a performance measure since an appropriate normalization method will reduce the variability of the normalized OTU abundances (raw counts divided by the size factor) due to different library sizes. A similar measure has been used in the evaluation of normalization performance for microarray data (*Fortin et al., 2014*). We use the traditional variance as the metric to assess inter-sample variability. For each method, the variance of the normalized abundance of each OTU across all samples is calculated and the median of the variances of all OTUs or stratified OTUs (based on their prevalence) is reported for each study. For each study, all methods are ranked based on these median variances. The distributions of their ranks across these 38 studies for each method are depicted in Fig. 6. A higher ranking (lower values in the box plot) indicates a better performance in terms of minimizing inter-sample variability.

In Fig. 6, we could see that GMPR achieves the best performance with top ranks in 22 out of 38 datasets, followed by CSS, which tops in seven datasets (Table S2). This result is consistent with the simulation studies, where GMPR and CSS are overall more robust to perturbations than other methods. Moreover, GMPR consistently performs the best for reducing the variability of OTUs at different prevalence levels. It is also noticeable that the inter-sample variability is the largest without normalization (RAW) and TSS does not perform well for a large number of studies. As expected, RLE only works for eight out of 38 datasets due to a large percentage of zero read counts. By adding pseudo-counts, RLE+ improves the performance significantly compared to RLE.

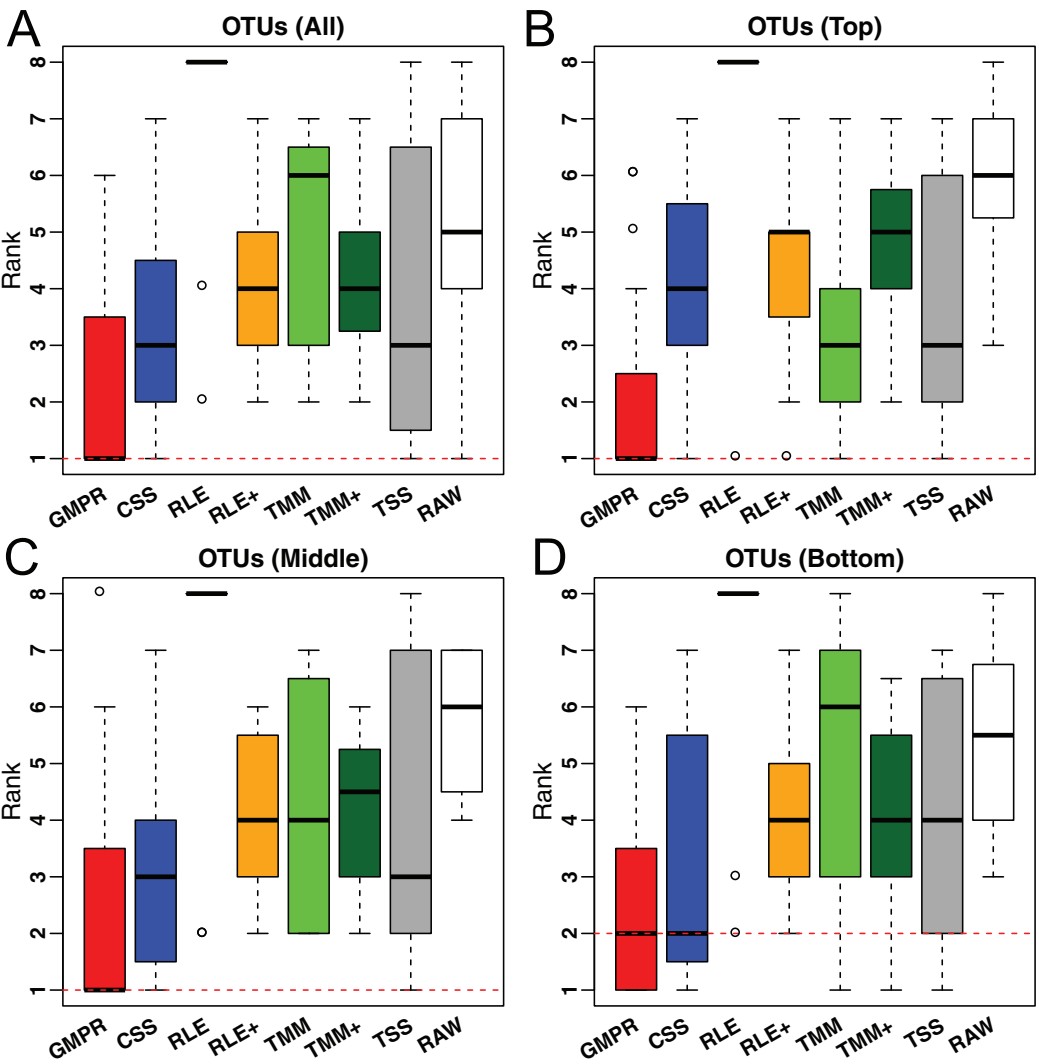

**Figure 6 Comparison of normalization methods in reducing inter-sample variability of normalized OTU abundances based on 38 gut microbiome datasets.** Distribution of the ranks for the medians of the OTU variances over the 38 datasets. The median is calculated over all OTUs (A) or OTUs of different prevalence level (B–D: Top, middle and bottom).

However, there is not much improvement of TMM+ compared to TMM. To see if the difference is significant, we performed paired Wilcoxon signed-rank tests between the ranks of the 38 datasets obtained by GMPR and by any other methods. GMPR achieves significantly better ranking than other methods ($P < 0.05$ for all OTUs or stratified OTUs). Fig. S1 compares the distributions of the OTU variances and their ranks for an example dataset (study ID 1561, all OTUs). Each OTU is ranked based on its variances among the competing methods. Although the difference in median variance is moderate, GMPR performs significantly better than other methods ($P < 0.05$, for all comparisons) and achieves a much lower rank.

To demonstrate the performance on low-diversity microbiome samples, we perform the same analyses on an oral and a skin microbiome dataset from Qiita (Table S1, bottom).

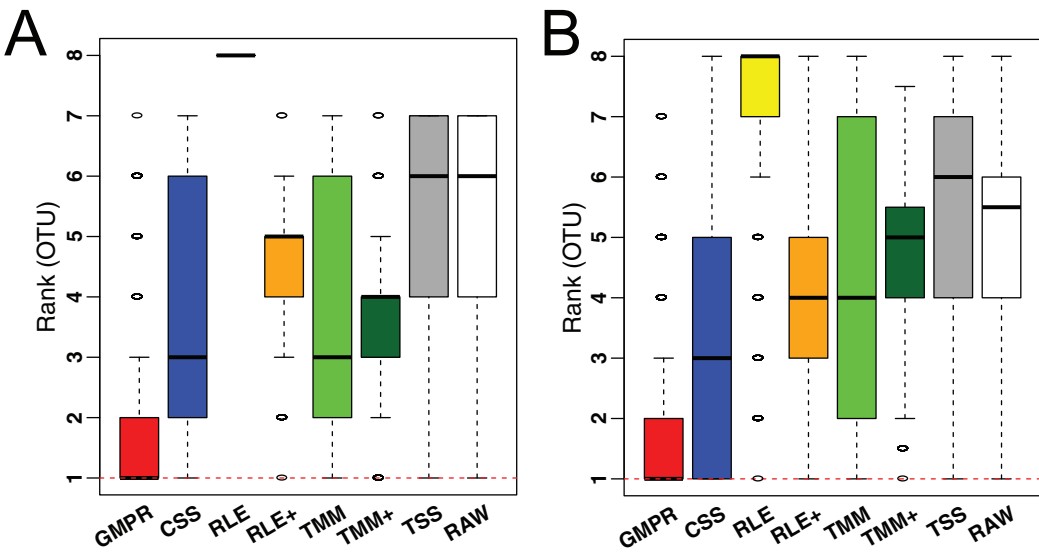

**Figure 7 Comparison of normalization methods in reducing inter-sample variability of normalized OTU abundances based on an oral (A) and a skin (B) microbiome dataset.** Distributions of the OTU ranks (each OTU is ranked based on its variances among the competing methods) are shown.

Consistent with the performance on gut microbiome datasets, although the difference in median variance is small (Fig. S2), GMPR achieves the lowest rank for majority of the OTUs, followed by CSS (Fig. 7).

## Real data: GMPR improves the reproducibility of normalized abundances

When replicates are available, we could evaluate the performance of normalization based on its ability to reduce between-replicate variability. Normalization will increase the reproducibility of the normalized OTU abundances. In this section, we compare the performance of different normalization methods based on a reproducibility analysis of a fecal stability study, which aims to compare the temporal stability of different stool collection methods (*Sinha et al., 2016*). In this study, 20 healthy volunteers provided the stool samples and these samples were subject to different treatment methods. The stool samples were then frozen immediately or after storage in ambient temperature for one or four days for the study of the stability of the microbiota. Each sample had two to three replicates for each condition and thus we could perform reproducibility analysis based on the replicate samples. We evaluate the reproducibility for the "no additive" treatment method for the data generated at the Knight Lab (*Sinha et al., 2016*), where the stool samples were left untreated. Under this condition, certain bacteria will grow in the ambient temperature with varying growth rates and we thus expect a lower agreement between replicates after four-day ambient temperature storage.

We conduct the reproducibility analysis on the core genera, which are present in more than 75% samples (a total of 26 genera are assessed). We first estimate the size

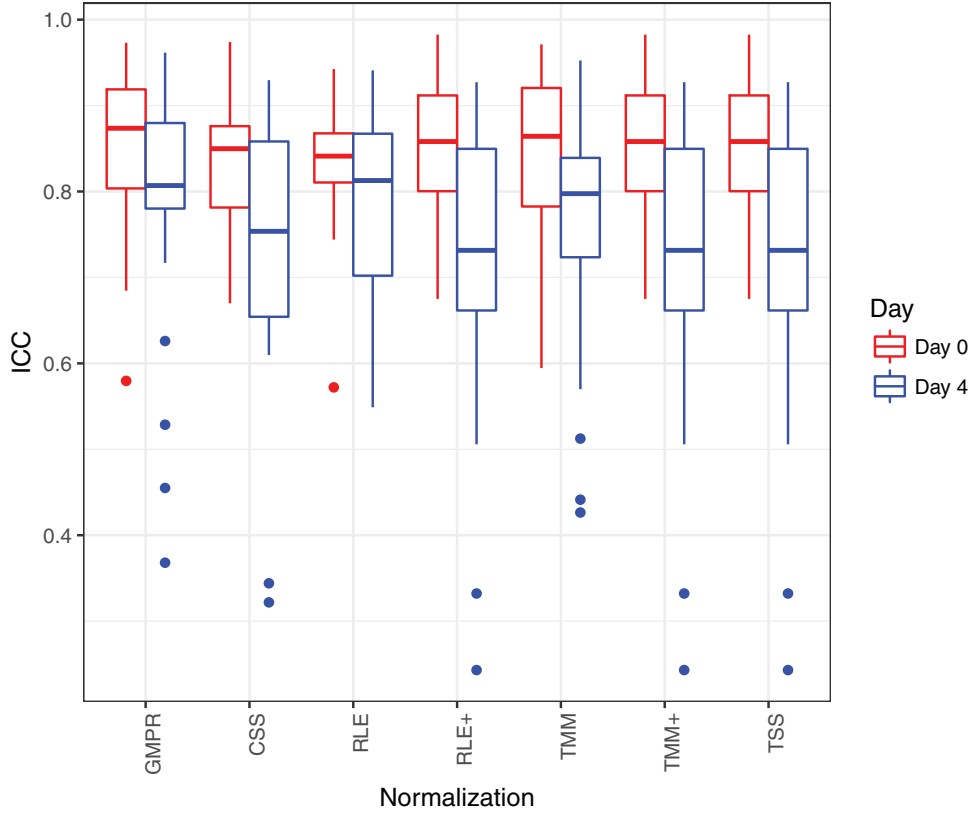

**Figure 8 ICC as a measurement for reproducibility is calculated for 26 core genera normalized by different methods for "day 0" and "day 4", respectively.**

factors based on the OTU-level data and the genus-level counts are divided by the size factors to produce normalized genus-level abundances. Intraclass correlation coefficients (ICC) is used to quantify the reproducibility for the genus-level normalized abundances. The ICC is defined as,

$$\rho = \frac{\sigma_b^2}{\sigma_b^2 + \sigma_\varepsilon^2}$$

where $\sigma_b^2$ represents the biological variability, i.e., sample-to-sample variability and $\sigma_\varepsilon^2$ represents the replicate-to-replicate variability. We calculate the ICC for 26 core genera for "day 0" (immediately frozen) and "day 4" (frozen after four-day storage), respectively. The ICCs are estimated using the R package "ICC" based on the mixed effects model. An ICC closer to one indicates excellent reproducibility.

Figure 8 shows that the reproducibility of the genera in "day 0" has higher reproducibility than "day 4" regardless of the normalization method used since reproducibility decreases as certain bacteria grow randomly as time elapses. While all the methods have resulted in comparable ICCs for "day 0," GMPR achieves higher ICCs for "day 4" than the rest methods. *Sinha et al. (2016)* showed that most taxa were relatively stable over four days and only a small group of taxa (mostly OTUs from *Gammaproteobacteria*) displayed a pronounced growth at ambient temperature. This

suggests that most of the genera may be temporally stable and their "day 4" ICCs should be close to the "day 0" ICCs. However, due to the compositional effect, if the data are not properly normalized, a few fast-growing bacteria will skew the relative abundances of other bacteria, leading to apparently lower ICCs for those stable genera. In contrast, the GMPR method is more robust to differential or outlier taxa as demonstrated by the simulation study, which explains higher ICCs for "day 4" samples.

## DISCUSSION AND CONCLUSION

Normalization is a critical step in processing microbiome data, rendering multiple samples comparable by removing the bias caused by variable sequencing depths. Normalization paves the way for the downstream analysis, especially for DAA of microbiome data, where proper normalization could reduce the false positive rates due to compositional effects. However, the characteristics of microbiome sequencing data, including over-dispersion and zero-inflation, make the normalization a non-trivial task.

In this study, we propose the GMPR method for normalizing microbiome sequencing data by addressing the zero-inflation. In one simulation study, we demonstrate GMPR's effectiveness by showing it performs better than other normalization methods in recovering the original library sizes when a subset of OTUs are differentially abundant or when random outlier OTUs exist. In another simulation study, GMPR yields better FDR control and higher power in detecting differentially abundant OTUs. In real data analysis, we show GMPR reduces the inter-sample variability and increases inter-replicate reproducibility of normalized taxa abundances. Overall, GMPR outperforms RNA-Seq normalization methods including TMM and RLE and modified TMM+ and RLE+. It also yields better performance than CSS, which is a normalization method specifically designed for microbiome data. As a general normalization method for zero-inflated sequencing data, GMPR could also be applied to other sequencing data with excessive zeros such as single-cell RNA-Seq data (*Vallejos et al., 2017*).

We note that the main application of GMPR method is for taxon-level analysis such as the presented DAA and reproducibility analysis, where it is important to distinguish those "truly" differential from "falsely" differential taxa due to compositional effects. Although we could apply the proposed normalization to (weighted) distance-based statistical methods such as ordination, clustering and PERMANOVA (*Caporaso et al., 2010*; *Chen et al., 2012*) based on the GMPR-normalized abundance data, simulations show that the advantage of using GMPR is very limited for such applications, compared to the traditionally used TSS method (i.e., proportion-based method) (Fig. S3). This is explained by the fact that the distance-based analysis focuses on the overall dissimilarity and the proportional data is already efficient enough to capture the overall dissimilarity. Probably, more important factors to consider in distance-based statistical methods are the selection of the most relevant distance measure and/or the application of appropriate transformation after normalization (*Costea et al., 2014*; *Thorsen et al., 2016*).

Besides the size factor-based approach (GMPR, CSS, TSS, RLE, TMM), the other popular approach for normalizing the microbiome data is through rarefaction. Both approaches have weakness and strength for particular applications. Although rarefaction discards a significant portion of the reads and is probably not optimal from an information perspective, it is still widely used for microbiome data analysis, particularly for α- and β-diversity analysis. The reason for its extensive use is that the majority of the taxa in the microbiota are of low abundance and their presence/absence strongly depends on the sequencing depth. Thus rarefaction puts the comparison of α- and β-diversity on an equal basis. Size factor-based normalization, on the other hand, is unable to address this problem. Thus rarefaction is still recommended for α- and β-diversity analysis, especially for unweighted measures and for confounded scenarios, where the sequencing depth correlates with the variable of interest (*Weiss et al., 2017*). For DAA, one major challenge is to address the compositional problem. Rarefaction has a limited ability in this regard since the total sum constraint still exists after rarefaction. In addition, it suffers from a great power loss due to the discard of a large number of reads (*McMurdie & Holmes, 2014*). In contrast, the size factor-based approaches are capable of capturing the invariant part of the taxa counts and address the compositional problem efficiently through normalization by the size factors. The size factors could be naturally included as offsets in count-based parametric models to address uneven sequencing depth (*Chen et al., 2018*).

Geometric mean of pairwise ratios is an inter-sample normalization method and has a computational complexity of $O(n^2 q)$, where $n$ and $q$ are the number of samples and features respectively. While GMPR calculates the size factors for a typical microbiome dataset ($n < 1,000$) in seconds, it does not scale linearly with the sample size. Large samples sizes are increasingly popular for epidemiological study and genetic association study of the microbiome (*Robinson, Brotman & Ravel, 2016*; *Hall, Tolonen & Xavier, 2017*), where tens or hundreds of thousands of samples will be collected to detect weak association signals. For such large sample sizes, GMPR may take a much longer time. A potential strategy for efficient computation under ultra-large sample sizes is to divide the dataset into overlapping blocks, calculate GMPR size factors on these blocks and unify the size factors through the overlapping samples between blocks. To increase the computational efficiency of GMPR for ultra-large sample sizes will be the focus of our future research.

### Funding
This work was supported by Mayo Clinic Center for Individualized Medicine. The funders had no role in study design, data collection and analysis, decision to publish, or preparation of the manuscript.

### Grant Disclosures
The following grant information was disclosed by the authors:
Mayo Clinic Center for Individualized Medicine.

## Competing Interests

Jun Chen is an Academic Editor for PeerJ.

## Author Contributions

- Li Chen analyzed the data, prepared figures and/or tables, authored or reviewed drafts of the paper, approved the final draft.
- James Reeve authored or reviewed drafts of the paper, approved the final draft.
- Lujun Zhang contributed reagents/materials/analysis tools, approved the final draft.
- Shengbing Huang prepared figures and/or tables, approved the final draft.
- Xuefeng Wang approved the final draft, offered expertise to improve the manuscript.
- Jun Chen conceived and designed the experiments, analyzed the data, prepared figures and/or tables, authored or reviewed drafts of the paper, approved the final draft.

## Data Availability

Github: https://github.com/jchen1981/GMPR

## Supplemental Information

Supplemental information for this article can be found online at http://dx.doi.org/10.7717/peerj.4600#supplemental-information.

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
