# Peer review of "GMPR: A robust normalization method for zero-inflated count data with application to microbiome sequencing data"

_PeerJ, doi:10.7717/peerj.4600_

## Round 0.1 · original submission · Major Revisions

The reviewers have raised concerns about benchmark results and have suggested additional analyses. Also, further discussion on comparing your methodology and rarefaction would improve the manuscript.

Reviewer 1 ·

Basic reporting

The paper by Chen et al. presents a novel method for normalizing micribiome read data for downstream analysis. The GMPR method seems like a good approach for normalization, and shows improved performance compared to current normalization methods. The paper is well written and provides good support for its claims.

Following are some comments which I feel can make the manuscript more clear and compelling, as well as explain better the reasons for the improved performance shown by GMPR:

1. While the GMPR method is presented with good details in the methods section, I am not convinced the reason for it performing better than the other methods it is compared to (RLE/TMM) is due to the higher sensitivity of these other methods to zeros in the data. As can be seen in figure 3, all methods (including GMPR) seem to degrade similarly as the fraction of 0 increases from 60% to 80%. I think it would benefit the paper to try and explain better the intuitive reason for the improved performance of GMPR.

2. Some observations are over/understated in the manuscript. specifically
- line 141-142 "the performance of TMM and RLE decreases significantly as the number of perturbed OTUs increases" - significantly is a bit too strong - it is a small change
- line 161-162 - "As shown in Figure 5A, although all approaches have slightly elevated FDRs relative to the nominal levels, the ".... slightly elevated is an understatement (a 5-fold underestimate in the FDR level for edgeR-GMPR and edgeR-TMM), please remove the "slightly"

3. Figure 4 and table 2 do not contribute a lot (since similar data is presented in figure 3 and figure 6 respectively). Maybe worth moving to supplementary file.

4. Some of the simulation details in the results section can be moved to the methods section instead (and then mentioned briefly in the results) in order to simplify the results section

5. The source code is available in the github address supplied. However, an additional file ("GMPR_0.1.3.tar.gz") is mentioned in the github readme file but is not there? where can it be obtained?

6. The paper is well written. Some small typos exist, such as:
methods section:
- c_ik and c_ki are interchanged sometimes (i.e. step1 and step2 use c_ik whereas definition and r_ij definition use c_ki
- Definition of S_i - can you describe with what S_i is used for (I assume it is the estimated number of reads in the sample, used for the normalization by division?)
- definition of r_ij has a multiplication over all j=1..n, which is not needed, as it is later performed in the S_i calculation
- formula for S_i calculation using r_ij - add ,i=1...n at end of formula (simiar to previous S_i calculation described to RLE).

results section:
line 125 - "trends to decrease" could be "decreases in most cases"
line 169 "sequencing of the stool samples" could be "sequencing of stool samples"
line 169 "These real datasets are retrieved from qiita database" could be "These experimental datasets were retrieved from the qiita database"
line 170 "with a sample size larger than 50" could be "with a sample size larger than 50 each
line 172 " from other body sites." change to " from other sample types"
line 215 "has achieved" change to "achieves"
line 232-233 "by showing its better performance" - change to "by showing it performs better"
line 243 " its use may not be limited to these applications." - change to " its use is not limited to these applications."

7. A few references can be added to the introduction when discussing compositionality effect (such as "Analysis of composition of microbiomes: a novel method for studying microbial composition" - Mandal et al. 2015, "Balance Trees Reveal Microbial Niche Differentiation" - Morton et al. 2017
Additionally, can you supply a reference to the statement in lines 50-51 "Many normalization methods have been developed for sequencing data generally, and for RNA-Seq data in particular..."

Box1 - maybe worth adding the reference for each method if possible.

Experimental design

The authors validate the performance of GMPS and compare to state of the art methods using both simulations and real datasets. A few issues which may help make the results and interpretation more robust:

8. For the real dataset used for the analysis shown in figure 6, were all scaling methods normalized somehow (i.e. sum/mean of all samples identical in all methods tested)? Otherwise, a normalization method that just divides all samples by a large number will obtain the lowest variance.
9. Can error bars be added to figures 3 and 4?
10. Can the performance of TSS be shown in figure 5? Or is there a technical reason preventing doing this?
11. For the real datasets mentioned in Table 1, were all samples from each study used? Some studies contain both fecal and non-fecal samples. Other studies contain fecal samples from multiple hosts (i.e. human and canine in study 3). Using all samples from each study may be problematic as the assumption of low variance may not hold for different hosts/sample types in the same study.
12. For the performance of the various methods shown in figure 6, can you also provide (maybe in the supplementary) an example of the distribution of the variance values for each method (such as a histogram of variance values) for some experiment from Table1? This can complement the comparison summary shown in figure 6 and enable seeing the difference between the methods.

Validity of the findings

13. In lines 207-208 "Under this condition, certain bacteria will grow in the ambient temperature and we thus expect a low agreement between replicates after four-day ambient temperature storage." - if growth happens similarly in all replicates of a given sample, would expect high ICCs? If so, maybe low ICCs are due to non-identical growth in the replicates maybe due to sporadic events? Or maybe ICC is between day 0 and day 4 (in that case need better explanation as i did not understand it this way).
Also same phrasing in lines 214-215 "since reproducibility decreases as certain bacteria grow as time elapses" - again is it due to growth which does not happen in all replicates?

14. In the conclusions section line 248 discusses usage with unweighted distance metrics. Is it possible to rarify after GMPR? will it improve performance over just rarefaction?

15. line 47-49 state "An ideal normalization method should thus capture the invariant part of the count distribution and be robust to outliers and differential features. The latter property is important to reduce the false positives due to compositionality." : Since going from the real bacterial abundances to the reads/sample is not a 1-1 function, the assumption used for inferring the abundances from the reads/sample (i.e. many biological relevant datasets have a large invariant part) should be clearly stated.

Additional comments

A few additional comments:
lines 34-35 " or more severely, result in falsely discovered features." - This can only happen when the library size is a dependent - so maybe combine with next sentence "Normalization is especially critical when the library size is a confounding factor that correlates with the variable of interest"

16. line 41-44 "First, it is not robust to outlier counts. Outliers have frequently been observed in sequencing samples due to technical artifacts such as preferential amplification by PCR (Aird et al., 2011). Several outliers could bias the library size estimates significantly." - what are outlier counts? how do they bias the library size estimates?

17. Lines 44-47 - why is the compositional effect enhanced when there is a small number of features ("...and the total number of features is relatively small...") - do you mean number of differentially abundant features?

18. line 62 - "One popular strategy to circumvent the zero inflation problem" - can you explain what the zero inflation problem is (i.e. why is having a large number of zeros problematic)?

Reviewer 2 ·

Basic reporting

The authors developed a new normalization method for microbiome amplicon data, which was extended from RLE. The usage of this method was demonstrated with both simulated and real data sets. It is good to see the statistic community is picking up the problem and start to solve the challenge in microbiome data set.

Experimental design

The current result looks good; however, there are several key issues that need to be addressed. Without these, I would be very skeptical about the general robustness and usefulness of GMPR method in microbiome:

1. In the simulation, the percentage of OTUs perturbed ranged from 0% - 32%. The authors should provide cases where over half of the OTUs are perturbed, because the authors tried to argue one of the advantage of GMPR, compared to other methods, is that it is robust even majority of the features are subject to change.

2. Besides stool microbiome data, the authors should demonstrate the usefulness of GMPR on skin and oral data, because skin and oral microbial community are much less diverse and thus has a bigger compositionality problem than stool data.

3. It is surprising that the authors didn't compare their method to rarefaction. As the most used normalization method, rarefaction has been shown to be applied for any type of analysis including quantitative or qualitative alpha/beta diversity, differential abundance, etc. The disadvantage of rarefaction is the loss of power due to the discard of reads when rarefying from high-read to low-read depths. The authors should include rarefaction for their benchmark results.

4. It looks like GMPR method could also be applied as a normalization method for quantitative beta diversity calculation (eg weighted UniFrac distance). The authors should show the benchmark results on this type of analysis as beta diversity is one of the most used techniques in microbiome data.

Validity of the findings

The data and analyses looks sound.

---

## Round 0.2 · Minor Revisions

I agree with the reviewers that the revised manuscript has improved substantially. However, the reviewers have raised a few minor issues and suggested additional presentation of data.

Reviewer 1 ·

Basic reporting

The manuscript convincingly presents the novel GMPR method for count normalization. GMPR seems to improve sensitivity and specificity compared to current methods in a large range of simulations and real sample types, and will help obtain better results in microbiome analysis.
The authors have addressed previous comments in a satisfactory manner.

A few small points:
1. In figures 3 and 4 for large perturbation, do the authors have an insight as to why TSS performance is increasing with the increase in proportion of perturbed OTUs (whereas as all other methods decrease)? If so, it would be useful to mention/explain in the corresponding results section.

2. When discussing figure 5, maybe worth explaining shortly why the observed FDR is higher than the nominal FDR even with the "balanced" scenario (If it is truly balanced, one would expect no compositional effect and hence full FDR control).

3. In general, can the authors comment on the optimal OTU filtering method to use prior to using GMPR? I would expect there is a balance between obtaining the ratios from low frequency OTUs (where discretization would play a large role - i.e. 2 reads in one sample and 1 read in another sample for the same OTU would be very noisy in estimating the fold change of the OTU between the two samples compared to 200 reads vs. 100 reads), whereas on the other hand keeping more OTUs would result in a larger number of ratios (and hence maybe a better estimation of the median). In the COMBO dataset, the authors filtered OTUs with <10 reads. Would this be the recommended filtering method? Or is the method not sensitive to the exact filtering?

4. Maybe worth adding in the introduction (where the COMBO dataset is first mentioned) a brief description of what the COMBO dataset is (i.e. human fecal etc.) for clarity to the reader without needing to read the reference?

4. typos:
line 153: missing space "zero inflated"
line 230: level should be levels

Experimental design

No additional comments

Validity of the findings

No additional comments

Additional comments

No additional comments

Reviewer 2 ·

Basic reporting

The authors improved the manuscript substantially and addressed all the comments. One figure I request the authors to add: besides showing the nonparametric ranks for inter-sample variability (fig 6&7), the actual value of the variances should be shown as well to see how much better or how different in terms of inter-sample variability among those normalization methods.

Experimental design

good

Validity of the findings

good

---

## Round 0.3 · accepted · Accept

Thank you for addressing all the comments by the reviewer. The paper has improved and the reviewers and I believe that GMPR will be an asset for future microbiome analyses.

# Reviewer 1 ·

Basic reporting

The authors have answered all points in a satisfactory manner.
I believe this paper will help future microbiome analysis, and GMPR seems an excellent method as a first step normalization prior to downstream analysis.

Experimental design

NA

Validity of the findings

NA

Additional comments

NA